# Dendritic Cells as a Therapeutic Strategy in Acute Myeloid Leukemia: Vaccines

**DOI:** 10.3390/vaccines12020165

**Published:** 2024-02-06

**Authors:** Francisca Palomares, Alejandra Pina, Hala Dakhaoui, Camila Leiva-Castro, Ana M. Munera-Rodriguez, Marta Cejudo-Guillen, Beatriz Granados, Gonzalo Alba, Consuelo Santa-Maria, Francisco Sobrino, Soledad Lopez-Enriquez

**Affiliations:** 1Department of Medical Biochemistry and Molecular Biology, and Immunology, School of Medicine, University of Seville, Av. Sanchez Pizjuan s/n, 41009 Seville, Spain; alepima97@gmail.com (A.P.); hala.bouita@hotmail.com (H.D.); camila.f.leivac@gmail.com (C.L.-C.); anamunrod2@alum.us.es (A.M.M.-R.); galbaj@us.es (G.A.); fsobrino@us.es (F.S.); 2Institute of Biomedicine of Seville (IBiS) HUVR/CSIC/University of Seville, Avda. Manuel Siurot s/n, 41013 Seville, Spain; mcejudo@us.es; 3Department of Pharmacology, Pediatry, and Radiology, School of Medicine, University of Seville, Av. Sanchez Pizjuan s/n, 41009 Seville, Spain; 4Distrito Sanitario de Atención Primaria Málaga, Sistema Sanitario Público de Andalucía, 29004 Malaga, Spain; beitagranados@hotmail.com; 5Department of Biochemistry and Molecular Biology, School of Pharmacy, University of Seville, 41012 Seville, Spain; csm@us.es

**Keywords:** personalized medicine, immunotherapy, mRNA, COVID-19, dendritic cells, vaccines

## Abstract

Dendritic cells (DCs) serve as professional antigen-presenting cells (APC) bridging innate and adaptive immunity, playing an essential role in triggering specific cellular and humoral responses against tumor and infectious antigens. Consequently, various DC-based antitumor therapeutic strategies have been developed, particularly vaccines, and have been intensively investigated specifically in the context of acute myeloid leukemia (AML). This hematological malignancy mainly affects the elderly population (those aged over 65), which usually presents a high rate of therapeutic failure and an unfavorable prognosis. In this review, we examine the current state of development and progress of vaccines in AML. The findings evidence the possible administration of DC-based vaccines as an adjuvant treatment in AML following initial therapy. Furthermore, the therapy demonstrates promising outcomes in preventing or delaying tumor relapse and exhibits synergistic effects when combined with other treatments during relapses or disease progression. On the other hand, the remarkable success observed with RNA vaccines for COVID-19, delivered in lipid nanoparticles, has revealed the efficacy and effectiveness of these types of vectors, prompting further exploration and their potential application in AML, as well as other neoplasms, loading them with tumor RNA.

## 1. Introduction

Dendritic cells (DCs) stand out as the most versatile professional antigen-presenting cells (APCs), which also include B cells and monocytes/macrophages, serving as a crucial link between innate and adaptive immunity, playing an essential role in triggering specific cellular and humoral responses against tumor and infectious antigens [1]. DCs are the only APCs able to interact with naive T cells, inducing their activation, differentiation, and polarized effector response [2]. Additionally, DCs play a fundamental role in immunological tolerance, making them a target for various therapeutic approaches, including antitumor therapies, autoimmune diseases, and hypersensitivity reactions. DCs represent a heterogeneous population with a controversial ontogeny (myeloid and lymphoid) [3,4], constituting approximately 1% or less of bone-marrow-derived cells in any lymphoid organ or the total cell count of blood and tissues.

Acute Myeloid Leukemia (AML) is a hematological malignancy characterized by the clonal and uninterrupted proliferation of aberrant myeloid stem cells due to genetic factors [5], such as somatic mutations in hematopoietic stem and progenitor cells with the capacity to self-renew and propagate the neoplastic clone. Some mutations, such as those in DNMT3A, TET2, and ASXL1, are commonly associated with early events in leukemia development, often observed in clonal hematopoiesis [6]. Similar to solid tumors, disease progression occurs despite endogenous immune responses against leukemic cells [7].

AML, a highly aggressive disease affecting primarily adults and having the shortest survival (5-year survival = 24%) [6], exhibits an increasing incidence in patients over the age of sixty-five [8,9]. Still, AML is fairly rare overall, accounting for only about 1% of all cancers. The impact of prognostic factors is intricately linked to the specific therapy administered to the patient, emphasizing the need for a tailored approach to the management of AML [10]. It can manifest de novo, follow previous treatment with chemotherapy, or result from myelodysplastic syndrome (MDS) or chronic myeloproliferative neoplasm. In the diagnostic assessment of suspected AML, a bone marrow aspirate is a routine procedure, with a marrow trephine biopsy considered optional unless a dry tap occurs; cytogenetic analysis and molecular testing are also used [10]. A diagnosis of AML typically requires a blast count of 20% or more in the marrow or blood, except for specific genetic subtypes and some erythroleukemia cases. Lineage involvement is often identified through cytochemistry, with myeloperoxidase (MPO) or Sudan black B (SBB) and nonspecific esterase stains (NSE) [10]. Initial treatment involves induction therapy based on chemotherapy, aiming to achieve complete remission (CR), followed by post-remission therapy to prevent relapse and eliminate neoplastic cells.

Beyond standard chemotherapy, which typically relies on anthracyclines and cytarabine [6], and radiotherapy treatment of neoplasms, immunotherapy (IT) emerges as an attractive and promising modality to stimulate patients’ immune systems, generally diminished [11,12,13]. In the same way, there is evidence that epigenetic modifications, specifically DNA methylation, are critical targets in hematopoietic malignancies, occurring at CpG within promoters, leading to the silencing of tumor suppressor genes, playing a pivotal role in cancer initiation and progression. Hypomethylating agents (HMAs) like azacitidine and decitabine have emerged as significant advancements in treating myeloid neoplasms, notably MDS and AML [6,14]. Thus, often employed as a second-line treatment after traditional therapies prove ineffective, DCs, referred to as ‘nature’s adjuvants’ [13], play a crucial role in enhancing antitumor immunity, being able to improve immunological response and clinical outcome [15]. DC-based cancer therapies encompass in vivo activation, in vivo expansion, blocking inhibitory signals, and vaccination [11,16,17,18,19,20]. Moreover, it is known as treatment-related acute myeloid leukemia (t-AML) when this pathology is generated as a consequence of the application of a cytotoxic chemical, radiation, or immunosuppression treatment prior to AML and not related to it. Recent studies indicate that t-AML represents 7–8% of total AML cases and the same as in the rest of the cases [21].

DC vaccines assume a significant role in preventing relapses and metastasis post-tumor excision or remission [2] while also contributing to disease progression stabilization in advanced stages. Thus, the objective of this review is to present the current status of antitumor vaccines based on DCs in the context of AML, as well as to highlight the novel approach of mRNA vaccines and their potential use in hematological malignancies.

## 2. Dendritic Cells: Ontogeny, Phenotypes, and Functions

DCs can be categorized based on their location, ontogeny, phenotype, and functions [22]. According to their ontogeny, they originate from common myeloid progenitors (CMPs). Depending on the presence or absence of transcription factors (TF), especially Nur77, they can modify themselves into two types of differentiated cells: (1) monocytes, under the influence of Nur77, transform into monocyte-derived dendritic cells (moDCs) under a pro-inflammatory microenvironment; (2) common DCs progenitors differentiate into plasmacytoid (pDCs) and conventional or myeloid DCs (cDCs) based on the action of essential TFs, such as Irf8 and E2-2 or Zeb2, Iraf4 and Iraf8, respectively. Moreover, cDCs can diverge into cDCs type 1 (cDC1) primarily under Irf8 and Batf3 influence and cDCs type 2 (cDC2) under Irf4 influence [11,23]. In this sense, the cDC2 population has been further subdivided into DC2 and DC3 or cDC2B, according to single-cell transcriptional profiles [24,25,26]. Moreover, cDC from the epidermis and dermis are known as Langerhans cells. On the other hand, a low proportion of pDC may arise from lymphoid progenitors, eventually giving birth to two functionally distinct subtypes of pDC with similar phenotypic characteristics [27,28,29,30].

DCs in peripheral tissue can exist in three activation states: immature (imDCs), semi-mature, and mature DCs (mDCs) [31], which present important phenotypical and functional differences and are usually associated with tolerogenic or immunogenic processes [32,33,34,35]. Upon recognizing external or internal danger signals, such as pathogen-associated molecular patterns (PAMPs) or danger-associated molecular patterns (DAMPs), DCs undergo activation [36]. Moreover, during this process, APCs migrate to lymphatic nodes to interact with T lymphocytes, inducing a specific effector response [37], with activation of intracellular signaling cascades, leading to the induction of a general pro-inflammatory response [36]. This transition from a resting state to activated DCs involves profound morphological, metabolic, phenotypical, and functional changes.

Functionally, noteworthy DC subsets in the context of tumors include cDC1, which is responsible for inducing cellular antitumor immunity and immunity against intracellular pathogens [2]. They express CD141, XCR1, CLEC9A, and DEC205 surface markers [2,23,26], correlating with a favorable prognosis in cancer [2]. They present the ability to migrate and infiltrate tumors. Moreover, cDC1 can release interferon (IFN)-III in response to toll-like receptors (TLR)-3 ligands [38], making this a good therapeutic strategy against tumors [39]. Furthermore, they cross-present tumor antigens to naïve CD8+ T lymphocytes in the lymph node and prime tumor-specific cytotoxic immune response by these cytotoxic T lymphocytes (CTLs), which is crucial for maintaining successful antitumor immunity. Moreover, complex interactions take place among cDC1 and innate cells such as macrophages, natural killer (NK), and natural killer T (NKT) cells to induce the interleukin (IL)-12 production from cDC1 in the tumor microenvironment. This is important for maintaining a T cell response during chemotherapy or immune checkpoint blockade. However, IL-12 production can be suppressed by IL-10 secreted by macrophages, tumor cells, and other immunosuppressive cells, as well as by tumor-derived factors, such as vascular endothelial growth factor (VEGF), that decrease cDC1 maturation. This is relevant to oncotherapy [11].

cDC2 is a more heterogeneous DCs subtype than cDC1, presenting CD141, CD11b, CD11c, CD1c, CD172a as main markers. They are responsible for promoting immunity mediated by CD4+ T lymphocytes in cancer, inducing activation of T helper (Th)17, Th1, and Th2 cells activation, as well as regulatory T (Treg) cells and CTLs, depending on the context [2]. Thus, as has been described above, cDC2 is nowadays dichotomized into DC2 and DC3. In this line, more studies are needed to determine whether both subtypes represent different stimulation stages or independent lineages [26]. However, Brown et al. proposed that DC3 can mediate Th17 pattern responses [24].

pDCs morphologically resemble plasma cells. They participate in the response to single-stranded viral RNA and DNA, as well as in the progression of autoimmune diseases. Regarding tumorigenesis, only myeloid pDCs seem to process and display antigens, and their presence is associated with a poor prognosis in solid tumors, such as breast and ovarian cancer [40,41]. However, Oshi et al. have shown that a high presence of pDC in triple-negative breast cancer (TNBC) was associated with better disease-specific and disease-free survival, suggesting the clinical relevance of pDC infiltration in TNBC [42]. pDCs activate CTLs through cross-presentation via the IFN-I pathway [2,11].

Contrary to cDCs, moDCs are less effective in transporting failures to lymph nodes and in the activation of T cells. They express CD14 and CD1a/CD1c surface markers, playing a key role in the inflammation process [43,44], and are often named “inflammatory mo-DC” [45]. However, they are the main source of DCs used in the development of DC-based vaccines. Thus, moDCs can promote T-cell differentiation but are poor inducers of CD4+ T-cell proliferation [46,47]. This might explain why extensive vaccination efforts using moDCs in cancer ITs have shown limited success so far, and alternatives using primary cDCs might be more promising [26,46,48]. However, moDCs induce activation in tumor-specific CD8+ T-cells [49]. Although their role in the development of spontaneous antitumor immunity remains to be elucidated, they are essential in maintaining immune responses during infections [11].

## 3. Dendritic Cells Maturation and Antitumor Immunity Induction

Three signals are needed to induce complete DC maturation and promote effective antitumor immunity [50]. Signal 1 is marked by up-regulation of major histocompatibility complex (MHC)/antigenic complexes on DCs, presenting them to T cell receptors (TCR) from CD8+ and CD4+ T lymphocytes, respectively, on the cell membrane. Thus, the acute intracellular increases will be maintained by MHC I, while phagocytosed exogenous elements will be presented via MHC II. Then, a positive balance between costimulatory molecules in DCs and their ligands, such as CD80-CD28, CD86-CD28, CD40-CD40L, OX40-and OX40L junctions, among others, promotes lymphocyte activation. Inhibitory interactions lead to the development of immune tolerance and tumor escape (signal 2). Finally, the production of cytokines and chemokines, such as IL-12 and IFN-I, as signal 3, promotes activation, differentiation, and immune memory of T cells [2,50,51], as illustrated in Figure 1.

Beyond the standard chemotherapy and radiotherapy treatment of neoplasms, IT is a very attractive and promising modality to stimulate patients’ immune systems, which are generally diminished due to immunosuppression caused by the tumor, achieving remission or stabilization of the disease [11,12,13]. Additionally, IT yields lasting responses from immunological memory [11]. Thus, IT is usually the second-line treatment when traditional main therapies have not been effective. DCs, designated as ‘nature’s adjuvants’ [13], are an interesting target in the immunotherapeutic field since they are potent inducers of antitumor immunity, improving the immunological response and clinical outcome [15]. In this sense, several studies have demonstrated that the presence of cDC1 in neoplasms is associated with a favorable prognosis [11], thus predicting the clinical outcome of melanoma patients and breast cancer [39,52].

## 4. Dendritic Cells-Based Therapies in Cancer: Vaccines

DC-based therapies in cancer include in vivo activation and expansion, blocking inhibitory signals mainly derived from the tumor microenvironment, which prevent DC infiltration and reduce their immunostimulatory activity, generating immunological tolerance and vaccination [11,16,17,18,19].

DC-based vaccines can be whole cell, peptide, or protein (free or bound to antibodies), viral vector-based, and mRNA or DNA vaccines [11,12]. While in whole cell, peptide, and genetic material vaccines, DC activation is carried out directly, in those that use viruses as vectors, it is carried out indirectly, activating DCs and the immune system once tumor cells are lysed due to the entry of the viral vector [11]. Among DC-based vaccines, those that include moDCs and leukemia-derived DCs (DCleu) are the most widely used [18]. Thus, DC vaccines play an essential role in preventing relapses and metastasis after tumor excision or remission [2], as well as promoting stabilization in terms of disease progression in its most advanced stages. Currently, a DCs vaccine, Sipuleucel-T (Provenge), for intravenous administration, is approved by the United States Food and Drug Administration (FDA), which has managed to extend the overall survival of patients with advanced prostate cancer resistant to hormone therapy for four months [2].

In contrast to the use of vaccines as monotherapy, several studies have demonstrated their potential synergy in tumor control when used in combination with both standard therapies and immune checkpoint inhibitors such as Programmed Death-ligand 1 (PD-L1) and CTL Antigen 4 (CTLA-4) [53].

The manufacture of DC vaccines mainly begins with the isolation of monocytes, CD34+ hematopoietic stem cells (HSC), and myeloid leukemic blasts to obtain imDCs or leukemia-derived DCs (DCleu) by adding granulocyte colony-stimulating factor- macrophages (GM-CSF) plus IL-4 or Prostaglandin E1 (PGE1) and Fms-related receptor tyrosine kinase 3 ligand (Flt3L), respectively. Then, once imDCs are obtained, they are ex vivo matured by the addition of a cocktail of several cytokines, such as TNF-α, prostaglandin-E1 (PGE1), prostaglandin-E2 (PGE2) and/or picibanil (OK-432) or CD40 ligand (CD40L) [54,55], and loaded with antigens through several pathways: whole cell antigens or tumor lysates, synthetic antigenic peptides, tumor DNA or mRNA, exosomes derived from cancer cells [54,56], Figure 2.

Therefore, by incorporating mRNA, it is possible to introduce not only antigens but also cell membrane proteins, such as CD40, CD70, and a constitutively active TLR-4 (known as TriMix) or TLR-4, CD40L, IFN-γ and dIL-10R-α (TetraMix) that modify the phenotype and facilitate or enhance DCs activation [56,57,58]. Moreover, mRNA-encoding telomerase, such as human telomerase reverse transcriptase (TERT), has been used to induce ex vivo mDCs, enhancing lysosomal targeting signal lysosomal-associated membrane protein (LAMP) to enhance immunostimulatory properties [59,60]. Lastly, mDCs are mainly administered to the patient intradermally or subcutaneously, although sometimes they are administered intravenously or intranodally [11], as shown in Figure 2.

DC vaccines are characterized by their clinical safety and feasibility [61,62,63]. Local reactions and flu-like symptoms predominate as side effects. Analytical alterations characterized by the appearance of antinuclear, antithyroid, and rheumatoid factor antibodies have been observed, but they do not develop serious autoimmune symptoms; in fact, these go from mild to none [56].

Vaccines made from peptides or proteins represent an alternative for active DC-based immunization against tumor antigens. The coupling of these antigenic peptides together with antibodies specifically directed to DCs protein receptors facilitates their targeting and improves the inoculated peptides’ presentation capacity, consequently activating these antigenic peptides. T lymphocytes are more attractive to the scientific community than free peptide antigen preparations [11,64] because the immune system has billions of different T lymphocytes, and each of them has different receptors but are arranged randomly. This repertoire represents a virtually infinite “drug library” for specific therapies that increase or decrease T cell function [32].

On the other hand, regarding vaccines based on viral vectors, Talimogene laherparepvec (T-VEC) stands out as a type of oncolytic IT (approved by the FDA) based on herpes simplex virus type I [65]. This virus has been attenuated and modified to replicate within the tumor cells, produce GM-CSF, and destroy neoplastic cells. The purpose of this strategy was to release a large proportion of antigens, which, together with the GM-CSF produced, will induce a systemic antitumor immune response. Its use is limited to advanced melanoma skin cancer [56,65].

Recently, nucleic acid-based vaccines such as DNA or mRNA have attracted great attention, as evidenced by the large number of newly published studies regarding them due to the spectacular results obtained with the vaccines against COVID-19 and against multiple solid tumors in several clinical trials [12]. It consists of the introduction of DNA or, generally, tumor naked mRNA or, preferably, through non-viral vehicles such as lipid nanoparticles (LNP), which will be captured by the DCs [66,67,68]. Once captured and processed, one or several antigens will be encoded to be presented in the MHC I/II complexes on their cell surface [12]. LNPs usually present functionalized molecules on their surface, with the aim of improving and directing their capture to DCs, which would present the targets on the outer membrane.

## 5. Dendritic Cells and Acute Myeloid Leukemia

AML is one of the neoplasms that benefit from DC-based therapies, specifically through the administration of the vaccines mentioned in the previous section.

AML is a blood cancer characterized by the uncontrolled growth of abnormal myeloid stem cells due to chromosomal changes or genetic mutations [5]. Similar to solid tumors, disease progression occurs despite endogenous immune responses against leukemic cells [7]. The accumulation of these cells in the bone marrow leads to the obliteration of normal hematopoiesis, triggering alterations in the red, white, and platelet blood series. In addition, patients experience other symptoms related to the metastatic invasion of myeloid blasts depending on the different organs of the body that they invade.

AML is an extremely aggressive condition, primarily affecting adults, and its incidence rises significantly among individuals aged sixty-five and older [8,9]. It can be generated de novo after previous treatment with chemotherapy or be the outcome of a MDS or a chronic myeloproliferative neoplasm. The treatment consists of an initial induction therapy based on chemotherapy, with the purpose of achieving CR, and post-remission therapy aimed to avoid relapse, eliminating the remaining chemoresistant neoplastic cells, known as measurable residual disease (MRD) or minimal disease. In the second phase of treatment, which includes chemotherapy and/or HSC transplantation (HSCT), a cytogenetic and molecular profile of the leukemia cells from the patient employed as response markers to therapy is essential [69].

Long-term overall survival is estimated at 25%, decreasing to <10% when the age group of patients is limited to ≥65 years [5]. This is due to the high probability of relapse in this group after initial treatment with chemotherapy, generally presenting a positive MME at the end of treatment, as they usually have and generate more aggressive genetic mutations that condition chemoresistance and tumor escape. Furthermore, a large proportion of these patients are not candidates for subsequent allogeneic HSCT (allo-HSCT) therapy due to the high morbidity and mortality that this practice entails. Currently, there is no agreed therapy in these cases in which patients have a high risk of relapse after remission but are not candidates for transplantation. Thus, although allo-HSCT is an option for many patients, it is not applied to everyone due to the high risk compared to other strategies or conditions in the patients themselves, which means that such treatment cannot be carried out. Among the therapeutic possibilities in these circumstances is IT, where DCs enter the scene. They have been considered a promising therapeutic option since they have the ability to interact and activate CD8+ T lymphocytes, NK, and NKT cells, which are mainly responsible for the elimination of neoplastic cells or as an adjunct to boost antigen-specific T cell ITs in AML [5,70,71,72].

Unlike other hematological malignancies, leukemia is the center of attention in terms of DC research, with DCs leading 70% of the clinical trials carried out [73]. This is a consequence of the advent of tyrosine kinase inhibitors, such as imatinib, in chronic myeloid leukemia (CML), where they are used as first-line treatment, representing a major turning point in response, survival, and mortality rates in this pathology. In fact, currently, it can be affirmed that CML patients treated chronically with these inhibitors have a life expectancy similar to the general population for subjects with similar age and gender. Similarly, acute lymphoblastic leukemia benefits from treatment with T-cell-directed and/or bispecific antibodies, which have significantly improved the prognosis of these patients. Due to the biological and cytogenetic characteristics of AML patients, generally >65 years, with complex genetic alterations, together with the lack of effective treatments in these patients, solved mostly in other types of leukemias, they become AML as a priority in the field of research, studying the possible use of IT, analogously to other hematological malignancies where the clinical benefits are evident.

### 5.1. Main Antigen Targets for Immunotherapy of AML

To facilitate the understanding of immune therapies and their potential synergism in AML, it is essential to expose the genetic, epigenetic, and phenotypic characteristics of myeloid blasts. Additionally, briefly describes their complex interactions with the environment where they reside as it conditions immune escape and immunosuppression.

AML is a highly heterogeneous entity, worsening as the disease progresses due to the appearance of new subclones that make it difficult to identify and select therapeutic targets. Antigen targets in this hematological neoplasia can be classified into three large groups: (A) Lineage-restricted antigens (LRAs). These antigens, such as CD123 and CD33, are present on tumor blast membranes and have the greatest clinical relevance. In this sense, CD33 is expressed in more than 90% of AML patients. Therapies against these antigens include the use of monoclonal antibodies or T cells modified in the expression of the chimeric antigen receptor (CAR-T). Currently, the only FDA-approved immunotherapeutic treatment available for this type of hematologic malignancy is an anti-CD33 monoclonal antibody combined with a drug, Gemtuzumab ozogamicin (GO). The main disadvantage of these targets and therapies is that these antigens are present, albeit to a lesser proportion, in HSC and extramedullary cells, causing hematological toxicity with their administration. This adverse effect can lead to treatment suspension [74,75]. (B) Leukemia-restricted antigens. These include wild-type proteins overexpressed in leukemia relative to normal cells, such as Wilms Tumor gene (WT1) and mucin1 (MUC1) or, conversely, poorly expressed proteins like TERT and survivin. There are antigens specifically located in certain types of tissues and cells, such as normal testicular germ cells or trophoblasts of the placenta, among others, such as Preferentially expressed Antigen in Melanoma (PRAME), Melanoma Antigen Gene (MAGE) and The CTA New York Esophageal Squamous Cell Carcinoma-1 (NY-ESO-1), also known as cancer-testis antigen 1B (CTAG1B). Most of these target peptides are found in intracellular compartments, a circumstance that requires their processing and presentation through the MHC for the generation of immune responses against them. WT1 participates in leukemogenesis, being less prone to immunological escape and, therefore, one of the most immunogenic antigens in AML. In fact, the measurement of WT1 expression levels in peripheral blood is used to detect post-chemotherapy MRD [74,76,77]. LRAs are the main source of antigens used in DC-based anti-AML vaccines. (C) Leukemia-specific antigens (LSA) are neoantigens or mutations in neoplastic cells, resulting in antigens classified as “non-self”, leading to greater immunogenicity and facilitating non-tolerance by the immune system. Examples include Nucleophosmin1 (NPM1), FMS-like tyrosine kinase 3 with internal tandem duplications (FLT3-ITD), Isocitrate Dehydrogenase 1 (IDH1), which generate gain of function or certain translocations such as the leukemic fusion proteins AML1-ETO, and PML-RARα, which give rise to expression fusion neoproteins, generally intracellular. AML is one of the neoplasms with the lowest mutation rate; however, a better quality is preferred over a higher mutational quantity to produce an effective antitumor response. Midostaurin, an anti-FLT3 drug, is currently available. This drug is administered together with induction chemotherapy in AML patients carrying this mutation. Its efficacy has been proven to increase survival from approximately 25 to 70 months [69,74].

Neoplastic cells reduce the expression of MHC I and II to decrease antigenic presentation to T lymphocytes. They also hinder and inhibit T cells through the overexpression of PD-L1 and Galectin-9 (Gal-9), among other mechanisms. The release of reactive oxygen species (ROS) into the bone marrow, where most tumor cells are located, suppresses T lymphocytes and, especially, NK cells, leading to specific lysis of target cells via the Fas receptor. Other released molecules, including arginase and indolamine 2,3-dioxygenase (IDO), promote the transformation of macrophages into their suppressor phenotype (M2), CD4+ and CD8+ effector T cells (Teff) apoptosis, and the induction of Treg to induce tolerance. Lastly, immunosuppressive immature myeloid cells (MDSCs) are activated through tumor-derived exosomes (EVs) to create an anti-inflammatory and immunosuppressive microenvironment with TGF-β, IL-10, and ROS production. Its presence in the bone marrow and peripheral blood correlates with MRD post-therapy [74,76].

The loss of antigens from harmful cells may result from the lack of presentation or non-expression of these proteins due to the methylation of the tumor DNA in AML neoplastic cells. Methylation causes non-transcription and translation, either by affecting the promoters or the coding genes themselves. This process is carried out by methyltransferases, which are involved in leukemogenesis, tumor proliferation, and evasion. Hypomethylating drugs, such as azatidine and decitabine, are therapeutic agents used in patients with high-risk AML (≥60 years), blocking genetic material methylation. This allows the expression of leukemia-associated antigens (LAAs) and previously inhibited tumor suppressor genes, triggering antileukemic responses and improving the efficacy of IT, such as anti-AML vaccines [76].

After chemotherapy administration, generally three days of anthracyclines plus seven days of cytarabine (3 + 7), and achieving a complete response, a consolidation treatment is applied to eradicate residual leukemia cells that could subsequently trigger a relapse of the illness. Except in the case of younger patients and those in the favorable prognosis group, most require HSCT. To guarantee the success of this practice, it is important that, at the time of the transplant, the patients do not present MRD, although its positivity would not contraindicate the procedure. To achieve this, prior to HSCT, a conditioning regimen is applied with cytostatic agents, suppressing the patient’s immune system and eliminating residual tumor cells [75].

As previously mentioned, most patients diagnosed with AML are over 65 years old, leading to the non-indication of transplantation due to the highly associated morbidity and mortality or the administration of less intense therapeutic regimens, although they allow a higher proportion of patients to benefit, the probability of response is lower [75].

The graft-versus-leukemia effect involves the removal of remaining neoplastic cells not eradicated by chemotherapy by the donor’s new cells, which recognize the remaining cells as foreign and initiate an immune response against neoplastic cells. This highlights that the immune system, induced by IT, can effectively deal with chemoresistant tumor clones, allowing disease control through stable remissions or control of the disease present [74].

As previously stated, IT applied to AML comprises checkpoint inhibitors (anti-PD-1, anti-PD-L1, anti-CTLA-4), T cells genetically redirected to leukemia cells (CAR-T therapies), antibodies against tumor antigens (GO, anti-CD33 antibodies), NK cell-based therapies, among others [78,79,80,81,82,83]. Additionally, other therapies, such as those based on anti-AML vaccines, are under clinical study [84,85,86].

In this context, DCs are cells capable of triggering strong and long-lasting adaptive immune responses, activating T lymphocytes while reinforcing the innate response through NK stimulation. Consequently, in AML, the primary immune soldiers are precisely the cells activated by DCs. Therefore, therapies based on the activation of DCs, both in vivo and in vitro, have been proposed and developed as a treatment against this leukemia, as exemplified by the anti-AML vaccines mentioned earlier.

### 5.2. DC Vaccines in AML

Vaccination strategies designed and applied in recent clinical trials are mainly divided into two groups: Autologous or allogeneic DC vaccines loaded with ex vivo AML antigenic peptides or RNA and Peptide or genetic material vaccines targeting DCs in vivo [87].

#### 5.2.1. Ex Vivo Loaded-DC Vaccines

(A) Autologous DCs. The autologous DCs used are derived from peripheral blood monocytes or stem cells extracted by leukapheresis in patients in remission. These cells, using standardized protocols cited above, differentiate into moDCs. Subsequently, their maturation and loading with LAAs are induced ex vivo to finally be reinfused into the patient. In this sense, there are multiple routes used to deliver antigens to these cells (tumor lysates, apoptotic tumor cells, exosomes derived from myeloid blasts, viral vectors, or direct inoculation of genetic material, proteins, and/or peptides), as illustrated in Figure 2 and detailed in Table 1.

It has been observed that the antigenic load of moDCs with tumor lysates and mRNA is higher than using these individually and separately in the context of T lymphocyte activation in vitro. This is based on findings obtained from several studies that compare the immune responses induced ex vivo depending on the number and variety of antigens inoculated into DCs [73], refer to Table 1.

Wang et al. designed an adenovirus capable of genetically modifying moDCs to encode a Suppressor of Cytokine Signaling 1 (SOCS1) RNA interference fragment (shRNA) as a checkpoint inhibitor, a survivin-MUC1 fusion protein, constituting the antigenic material, and a bacterial flagellin, TLR-5 agonist that contributes to DC maturation. Higher immunogenicity was observed in DCs treated with the viral vector compared to the control, with an increase in the production of CTL and survivin- and MUC1-specific CD4+ T lymphocytes. The vaccines, likewise, were safe and well tolerated by AML patients [88]. Other researchers introduced WT1 mRNA into moDCs of the 30 AML patients who received the vaccine without simultaneous chemotherapy treatment, and 13 (43%) patients presented an antileukemic response [70]. Moreover, the loading not exclusively of WT1 mRNA in DCs but of other antigens has also shown clinical benefit. The results of a phase I trial led by the Lichtenegger et al. team were promising; they introduced the mRNA of the WT1 and PRAME antigens and CMVpp65 into DCs, obtaining their maturation through TLR7/8 [90] (Clinicaltrials.gov NCT01734304; Table 2).

Chevallier P. et al. conducted a phase I/II trial using a DCs vaccine pulsed with autologous leukemic apoptotic cells in elderly AML patients in the first or second complete remission. Five autologous DC vaccines were manufactured and injected on days +1, +7, +14, +21, and +35. This approach demonstrated higher immunogenicity and adaptability to each patient, independently of the tumor antigens type [87] (Clinicaltrials.gov NCT01146262; Table 2).

Recently, Floisand et al. utilized FDC101, an autologous RNA-loaded mDCs vaccine expressing both WT1 and PRAME LAAs. They proposed FDC101 as maintenance therapy for AML patients in CR1. These DC-vaccines were injected for two years to cover the main risk period for relapse in AML patients who achieve CR1 with intensive chemotherapy but are ineligible for allo-HSCT [72] (Clinicaltrials.gov NCT02405338; Table 2).

In addition to moDCs, other methods have been considered for the generation of immunogenic DCs, such as DCs derived from leukemic blasts (AML-DCs), which are produced from neoplastic blasts. For their generation, in the first phase, blast cells are obtained from peripheral blood or bone marrow and cultured with GM-CSF, IFN-α, IL-4, and FLT3-L, leading to differentiation and obtaining imDCs. After PAMP stimulation, mDCs are obtained. Unlike moDCs, AML-DCs do not require antigenic loading since they have the entire catalog of AML antigens, generating immune responses that are not limited to a single antigen. Workgroups experimenting with this vaccine evidenced an antileukemic response, with the production of CTL and CD4+ T lymphocytes specific to WT1 and PRAME. However, no clinical benefit was demonstrated despite initial disease control. Furthermore, after direct comparison with moDCs loaded with ex vivo antigens, a lower efficiency in mobilizing antitumor lymphocytes was observed. Possible reasons for this, although not proven yet, could be the absence of the 4-1 BB ligand, necessary and relevant in costimulation, and the expression of IDO-1, leading to immune tolerance, by AML-DCs [5,74,92].

On the other hand, unlike AML-DCs, hybridomas are the outcome resulting from the fusion of leukemic myeloid blasts with the patient’s DCs, generally derived from monocytes. Similar to AML-DCs, ex vivo antigenic loading is not necessary. In a trial, 17 vaccinated patients with hybridomas who were in remission after chemotherapy demonstrated long-term remission in 12 (71%) of cases, with a median follow-up of 57 months. Disease control coincided with the increase and lasting expansion of leukemia-specific CD8+ and CD4+ T lymphocytes, specifically WT1, MUC1, and NY-ESO antigens in patients with HLA-A *0201 [93], Clinicaltrials.gov NCT01096602, Table 2.

Finally, regarding autologous pDCs, although there are therapeutic advances for the use of pDCs-based vaccines in cancer therapy, pDCs have not been used exclusively in therapies for AML [94].

(B) Allogeneic DCs. Van de Loosdrecht et al. generated the allogeneic DC vaccine DCP-001, obtained from a cell line derived from AML, which expresses a wide variety of LAAs. It was administered to 12 elderly AML patients who were at high risk of relapse and were not candidates for standard post-remission therapies [95]. The results showed an antileukemic response superior to that achieved with autologous cells since antitumor immunity is reinforced by the activation of Th1 lymphocytes in response to alloantigens. As a result of the information obtained in this phase I trial, a randomized, multicenter phase II trial was initiated to obtain more consistent data [74,95] (Clinicaltrials.gov NCT03697707; Table 2).

In a manner analogous to blast cells, some researchers used allogeneic moDCs from healthy donors, into which WT1 peptides are introduced or loaded with lysates derived from leukemic cells [73]. Currently, strategies focused on obtaining cDCs, mainly cDC1, in vitro from allogeneic CD34+ stem cells, allowing the inoculation of these against the moDCs, which migrate to the lymphoid organs, where they present the antigens. The use of allogeneic cells ensures the supply of DCs in sufficient quantities and without functional disorders, observed effects, and side effects to the disease and its treatment [73].

#### 5.2.2. In Vivo DC-Targeting Molecules Vaccine Carrier

Three approaches are considered in the activation of immunization, such as targeted antibodies fused with antigens, organic molecules like proteins/peptides or mRNA, and ionizable lipid nanoparticles (LNPs) and virus-like particles (VLPs).

The targeted antibody strategy involves monoclonal IgG isotypes that bind to different antigens (mainly NY-ESO 1 and WT1) immunoglobulins (Ig), specifically targeting receptors on DCs. Generally, they are administered with adjuvants to promote immunogenicity and avoid tumor-associated immunosuppression. In addition, point mutations are made in the heavy and light chains of the inoculated antibodies to promote their stability and success as a therapeutic agent [77].

The target receptors studied in DCs include some C-type lectin receptors (CLR), such as DEC-205 or CD205, CLEC9A (DNGR-1), DC-SING, or chemokine receptors, particularly CXCR1.

Anti-CD205 antibodies conjugated to the NY-ESO-1 antigen have shown to be useful in patients with malignant solid tumors, inducing antigen-specific antileukemic responses according to the results of a phase I clinical trial [96]. Similar results were reported in another phase I trial where seven patients with AML or MDS (a condition that precedes AML in many cases) were administered CDX-1401 (anti-DEC205 and NY-ESO-1), followed by several cycles of decitabine [91], Table 2.

Pearson et al. utilized the CLEC9A receptor (DNGR-1), which is exclusively located in cDC1. They compared the immunogenicity and clinical response rates of three vaccines consisting of the fusion of WT1 with an anti-CLEC9A, anti-DEC205, or anti-β-galactose antibody, with anti-CLEC9A-WT1 antibodies showing high reactivity in a humanized mouse model [77].

While carriers delivering antigens based on VLPs have not yet been used in humans, organic molecules (penetrating cationic polymers and peptides) and NPLs have undergone testing. Thus, it was shown that the inclusion of mannose in the membrane of LNPs improved their uptake by DCs through the mannose receptor CD206 [97]. However, Zhang et al. demonstrated the presence of an immune response using biodegradable polymers derived from lactic and glycolic acids (PGLA) containing the WT1 antigen in AML patients [98].

## 6. Conclusions

Antitumor vaccines present a promising horizon in AML, potentially advancing us toward the long-awaited era of personalized medicine. DC-based vaccines enhance immunogenic capacity and provide clinical benefits. In vitro studies support the therapeutic potential of checkpoint inhibitors in conjunction with anti-AML immunizations. Additionally, another emerging and promising strategy involves combining these vaccines together with CAR-T cell therapy. AML-specific monoclonal antibodies also synergize with vaccination therapy since antibody-induced cellular cytotoxicity triggers the release of antigens. However, no clinical studies have been developed employing this therapeutic strategy. While LNPs with tumor mRNA have been studied so far exclusively in certain solid tumors, it would be interesting to expand the field of action to include hematological neoplasms among the subsidiaries of this antitumor therapeutic novelty. Future directions in therapeutic approaches for patients with AML should consider the progress made, given the necessity for highly personalized therapies. The consolidation of the use of autologous or allogeneic whole tumor cell lysates and/or sets of tumor-derived peptides as a source of tumor-associated antigens for loading DCs reduces the possibility of immune escape after vaccination. All of the factors, coupled with the rise of mRNA platforms, encourage optimism towards developing a personalized vaccine for these patients.

## Figures and Tables

**Figure 1 vaccines-12-00165-f001:**
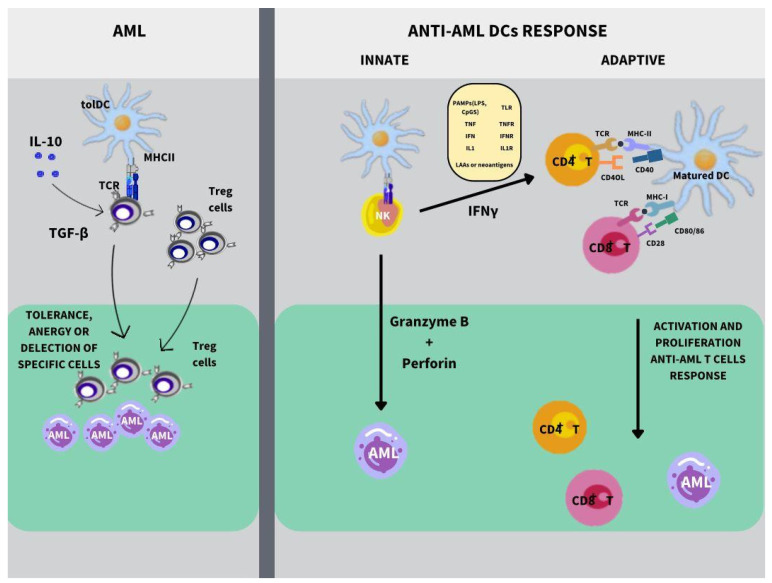
Role of DCs in AML. DCs are downregulated in AML, which are called tolerogenic DCs (tolDCs), and play an important role in inducing peripheral tolerance, anergy, or deletion of T cell clones via the activation of Treg cells. Contrary to DCs under inflammatory stimuli such as TLR ligands, LAAs can stimulate both innate and adaptive immune responses against AML cells. DCs and NK cells, as innate cells, interact in a reciprocal way to induce rapid and potent anti-AML responses by realizing cytolytic granules, such as granzyme B and perforin. Moreover, NK cells, after interacting with DCs, produce IFN-γ, serving as a bridge between innate and adaptive immunity, promoting the activation, proliferation, and anti-AML CD4 and CD8 T cells response.

**Figure 2 vaccines-12-00165-f002:**
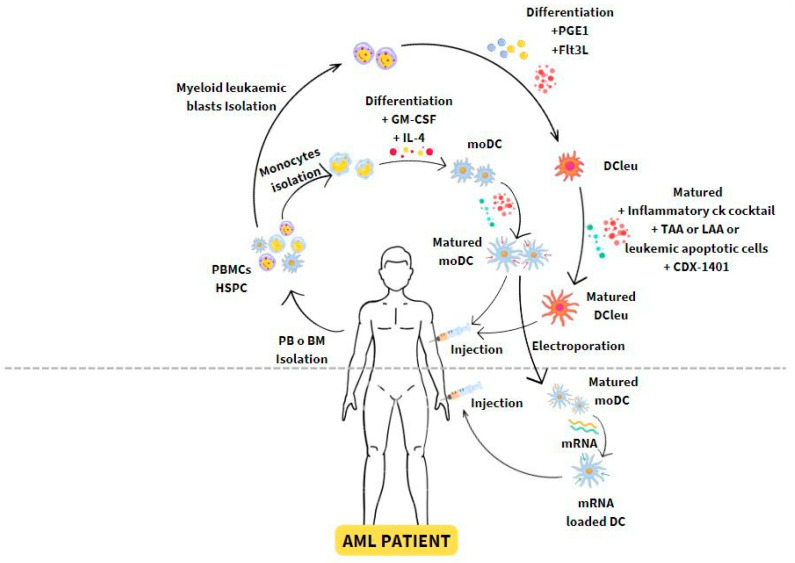
DC vaccination as an immunosuppression strategy in AML. Immature DCs or monocytes are extracted from peripheral blood samples, isolated, and loaded with tumor leukemia-associated antigens (LAAs) or neoantigens. The mDCs are then injected back into the same patient to induce tumor-specific effector T-cell responses.

**Table 1 vaccines-12-00165-t001:** Molecules for pulsed or loaded DC-vaccines for AML.

Molecules	Delivery Methods	Achievements	References
hTERT	intradermal	hTERT-DCs ↑ survival among AML patients.	[60]
survivin-MUC1+flagelin	injection	SOCS1-silenced DCs ↑ safety and efficacy in treatment of relapsed AML.	[88]
WT1 mRNA	intradermal	Clinical response correlated with ↑ of WT1-specific CD8 T circulating cells.	[70,89]
WT1, PRAME, and CMVpp65 mRNA	intradermal	induction of leukaemia-specific primary and secundary immune responses.	[90]
leukemic apoptotic cells	intradermal	↑ Median survival to double what was expected in AML patients.	[87]
WT1, PRAME mRNA	intradermal	induce antigen-specific T cell responses in AML patients.	[72]
CDX-1401 (anti-DEC205 and NY-ESO-1)	subcutaneous and intradermal	an antigen-specific immune response.	[91]

↑: Increase.

**Table 2 vaccines-12-00165-t002:** Summary of the most recent clinical trials of DC-based vaccines in AML.

DCs Type	Phase	LAAs	CODE
Autologous TLR7/8-matured DCs	I	TLR7/8-matured DCs + mRNA-transfected encoding WT1, PRAME and CMVpp65	NCT01734304
Autologous DCs	I/II	Leukemic apoptotic corpse	NCT01146262
Autologous DCs	I/II	mRNA-transfected encoding WT1 and PRAME	NCT02405338
Autologous DCs/AML fusion cells	II	AML blasts	NCT01096602
DCs +AML fusion cells	II	AML blasts	NCT03059485
DCs	I	Overlapping peptides mixes derived from full-length NY-ESO-1, MAGE-A1, and MAGE-A3	NCT01483274
Autologous or HLA-matched donor-derived DCs	I	Expressing WT1/hTERT/Survivin	NCT05000801
Autologous DCs	I	WT1 mRNA-transfected	NCT00834002
Autologous DCs	II	GRNVAC1 transfected with mRNA encoding hTERT	NCT00510133
Autologous DC + leukemic fusion cells	I	AML blasts	NCT00100971
Autologous DCs	II	Electroporated with WT1 mRNA	NCT01686334
Autologous DCs	I-II	Electroporated with WT1 mRNA	NCT03083054
DCs + autologous leukemic cells	I	STING-Dependent Activators (STAVs) loaded autologous leukemic cells	NCT05321940
Allogeneic DCs	II	DCP-001	NCT03697707
Allogeneic DCs	I-II	WT1 peptide-loaded	NCT00923910

## Data Availability

Not applicable.

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
