# Peer review of "Dendritic Cells as a Therapeutic Strategy in Acute Myeloid Leukemia: Vaccines"

_vaccines, 2024, doi:10.3390/vaccines12020165_

Round 1
Reviewer 1 Report
Comments and Suggestions for Authors Type of the Paper (ArticleTitle: Dendritic Cells as a Therapeutic Strategy in Acute Myeloid Leukemia: Vaccines
Journal: Vaccines MDPI
Considering the importance of the topic, i.e., the development of novel therapeutic approaches for the treatment of Acute Myeloid leukemia, as well as the high quality of the review, the manuscript is relevant and interesting. The manuscript is in general well written, while the scientific writing is good. In this review manuscript, the authors described the currently published knowledge behind the usage of dendritic cells as a novel therapeutic approach as an adjuvant treatment in AML following initial therapy. In general, this approach demonstrated promising outcomes in preventing or delaying tumor relapse and exhibits synergistic effects when combined with other approaches, so the importance of using of dendritic cells-based vaccines for treating Acute Myeloid Leukemia is relevant. The vast literature in this interesting topic is adequately covered. Both figure 1 and table 1 are well designed. Considering some minor improvements, I recommend a minor revision. My comments/observations are as follows:
My main comment is to improve the introduction. I suggest including a brief overview on Acute Myeloid Leukemia, including epidemiology, diagnosis, and prognosis with particular attention on the most effective therapeutic approaches currently employed in the clinic as well as the most promising treatments with are currently in the pre-clinical setting. For instance, there is a vast literature on the use of epigenetic based drugs for treating AML, they deserve attention and should be mentioned. These are important notions which should be included in the work in order to improve its completeness of information. Authors can check and mention
a. https://pubmed.ncbi.nlm.nih.gov/35101585/
b. https://www.nature.com/articles/s41375-021-01218-0
c. https://ashpublications.org/blood/article/115/3/453/27145/Diagnosis-and-management-of-acute-myeloid-leukemia
d. https://www.annalsofoncology.org/article/S0923-7534(20)36079-8/fulltext
e. https://ashpublications.org/blood/article/140/12/1345/485817/Diagnosis-and-management-of-AML-in-adults-2022
MINOR OBSERVATIONS
1. 1.Please revise several typo errors
2. Bold worlds should be avoided
3. If conducted, clinical trials relying on dendritic cells-based vaccines should be mentioned, while codes listed, for instance in a novel table4. An addtional figure describing the main functions of DC cells even in the context of AML would improve the quality of the work
Author Response
Point-by-point Reply
Reviewer #1
Considering the importance of the topic, i.e., the development of novel therapeutic approaches for the treatment of Acute Myeloid leukemia, as well as the high quality of the review, the manuscript is relevant and interesting. The manuscript is in general well written, while the scientific writing is good. In this review manuscript, the authors described the currently published knowledge behind the usage of dendritic cells as a novel therapeutic approach as an adjuvant treatment in AML following initial therapy. In general, this approach demonstrated promising outcomes in preventing or delaying tumor relapse and exhibits synergistic effects when combined with other approaches, so the importance of using of dendritic cells-based vaccines for treating Acute Myeloid Leukemia is relevant. The vast literature in this interesting topic is adequately covered. Both figure 1 and table 1 are well designed. Considering some minor improvements, I recommend a minor revision. My comments/observations are as follows:
My main comment is to improve the introduction. I suggest including a brief overview on Acute Myeloid Leukemia, including epidemiology, diagnosis, and prognosis with particular attention on the most effective therapeutic approaches currently employed in the clinic as well as the most promising treatments with are currently in the pre-clinical setting. For instance, there is a vast literature on the use of epigenetic based drugs for treating AML, they deserve attention and should be mentioned. These are important notions which should be included in the work in order to improve its completeness of information. Authors can check and mention
- https://pubmed.ncbi.nlm.nih.gov/35101585/
- https://www.nature.com/articles/s41375-021-01218-0
- https://ashpublications.org/blood/article/115/3/453/27145/Diagnosis-and-management-of-acute-myeloid-leukemia
- https://www.annalsofoncology.org/article/S0923-7534(20)36079-8/fulltext
- https://ashpublications.org/blood/article/140/12/1345/485817/Diagnosis-and-management-of-AML-in-adults-2022
We greatly appreciate the accurate and detailed review provided by the referee, and we concur with the observations made by Reviewer #1. Although part of his/her proposal was included in subsection 5. (lines 282-316), we have included following their recommendations a brief overview of Acute Myeloid Leukemia in the introduction section Furhtermore, we have diligently included the references that you have kindly and accurately proposed to us.
MINOR OBSERVATIONS
- 1.Please revise several typo errors
- Bold worlds should be avoided
Thank you very much for your appreciation. We have addressed the identified errors and the use of bold formatting has been omitted from the text.
- If conducted, clinical trials relying on dendritic cells-based vaccines should be mentioned, while codes listed, for instance in a novel table
We greatly appreciate this particular reviewer’s comment, and we concur with the reviewer’s assessment. In response, we have introduced a new table, designated as Table 2. Summary of the most recent clinical trials DCs-based vaccines in AML.
- An addtional figure describing the main functions of DC cells even in the context of AML would improve the quality of the work
We express our gratitude to Reviewer #1 for his/her appropriate suggestion, and we have included a new figure, denoted as Figure 1. Role of DCs in AML, into the text.

Reviewer 2 Report
Comments and Suggestions for Authors
The work expresses the topic with a great deal of detail, well explaining the starting background and listing all the various therapeutic possibilities present in the literature, also explaining them in an articulated way and with examples.
The treatment of all immunotherapies involving dendritic cells and possible therapeutic associations is impeccable.
The only flaw that I believe can be found in the paper is that it is a list of what exists in the field, without however giving the author's priority on which is the path that shows the greatest possibilities for development and which is the path for the author to follow for the future. All paths are put on the same level while it would be more authoritative to give the author's vision of the near therapeutic future.
I believe it is right to improve the conclusions by summarizing the entire panorama exposed but giving what the author believes to be the direction to follow.
Author Response
Point-by-point Reply
Reviewer #2
The work expresses the topic with a great deal of detail, well explaining the starting background and listing all the various therapeutic possibilities present in the literature, also explaining them in an articulated way and with examples.
The treatment of all immunotherapies involving dendritic cells and possible therapeutic associations is impeccable.
The only flaw that I believe can be found in the paper is that it is a list of what exists in the field, without however giving the author's priority on which is the path that shows the greatest possibilities for development and which is the path for the author to follow for the future. All paths are put on the same level while it would be more authoritative to give the author's vision of the near therapeutic future.
I believe it is right to improve the conclusions by summarizing the entire panorama exposed but giving what the author believes to be the direction to follow.
We thank the kindly reviewer’s comments. We highly appreciate for his/her accurate and appropriate suggestion, acknowledging the valididty of the reviewer’s. In response, we have improved the conclusions section by incorporating our perspective of the near future in immunotherapy and AML, outlining the direction to be pursued.

Reviewer 3 Report
Comments and Suggestions for Authors
The work analyzes very well the current state of the art of vaccination practices in AML, even if it is an approach that is further away from the patient's bedside than we can imagine in the paper. It is interesting to integrate with literature data, if present, regarding vaccination practices in plasmacytoid dendritic cell AML.
Author Response
Point-by-point Reply
Reviewer #3
The work analyzes very well the current state of the art of vaccination practices in AML, even if it is an approach that is further away from the patient's bedside than we can imagine in the paper. It is interesting to integrate with literature data, if present, regarding vaccination practices in plasmacytoid dendritic cell AML.
We thank the kindly reviewer’s comments. We sincerely appreciate for his/her accurate and appropriate suggestion, acknowledging the correctness of the reviewer’s assessment. Thus, we have reviewed the literature specifically focusing on the utilization of pDCs-based vaccines in AML, and although there are therapeutic advances in pDCs for cancer treatment [1], we have not located their use in this type of cancer specifically.
In this sense, following your wise suggestion, we have included this appreciation in the text, lines 496-498.
REFERENCE
- Hernandez, S.S.; Jakobsen, M.R.; Bak, R.O. Plasmacytoid Dendritic Cells as a Novel Cell-Based Cancer Immunotherapy. International journal of molecular sciences 2022, 23, doi:10.3390/ijms231911397.
